# CRASH: Crash Recognition and Anticipation System Harnessing with Context-Aware and Temporal Focus Attentions

## ABSTRACT

Accurately and promptly predicting accidents among surrounding traffic agents from camera footage is crucial for the safety of autonomous vehicles (AVs). This task presents substantial challenges stemming from the unpredictable nature of traffic accidents, their long-tail distribution, the intricacies of traffic scene dynamics, and the inherently constrained field of vision of onboard cameras. To address these challenges, this study introduces a novel accident anticipation framework for AVs, termed CRASH. It seamlessly integrates five components: object detector, feature extractor, object-aware module, context-aware module, and multi-layer fusion. Specifically, we develop the object-aware module to prioritize high-risk objects in complex and ambiguous environments by calculating the spatial-temporal relationships between traffic agents. In parallel, the context-aware is also devised to extend global visual information from the temporal to the frequency domain using the Fast Fourier Transform (FFT) and capture fine-grained visual features of potential objects and broader context cues within traffic scenes. To capture a wider range of visual cues, we further propose a multi-layer fusion that dynamically computes the temporal dependencies between different scenes and iteratively updates the correlations between different visual features for accurate and timely accident prediction. Evaluated on real-world datasets—Dashcam Accident Dataset (DAD), Car Crash Dataset (CCD), and AnAn Accident Detection (A3D) datasets—our model surpasses existing top baselines in critical evaluation metrics like Average Precision (AP) and mean Time-To-Accident (mTTA). Importantly, its robustness and adaptability are particularly evident in challenging driving scenarios with missing or limited training data, demonstrating significant potential for application in real-world autonomous driving systems.

## CCS CONCEPTS

• **Applied computing** → **Physical sciences and engineering**.

## KEYWORDS

Traffic Accident Anticipation; Autonomous Driving; Spatial-Temporal Analysis; Fast Fourier Transform; Dynamic Visual Fusion

## 1 INTRODUCTION

The introduction of Advanced Driver Assistance Systems (ADAS) and Autonomous Vehicles (AVs) marks a significant leap forward in our quest for safer roads [1, 13, 19]. By aiming to predict and prevent traffic accidents before they happen, these technologies are at the forefront of transforming our transportation landscape. This capability is crucial, enabling vehicles to make decisions that avoid collisions and protect passengers [6].

Despite the progress we have made, the road to reliable accident anticipation is filled with hurdles. Traffic, by nature, is chaotic and full of surprises. From a sudden stop in the flow to a pedestrian stepping out unexpectedly, the variables are endless. This complexity is compounded when you consider the diversity of how accidents can occur, the subtle yet vital visual cues that can get lost among everyday traffic elements, and the unpredictable behavior of other road users. The current solutions are inadequate in several ways when they come to addressing these issues:

**Firstly**, existing methods are predominantly object-centric, relying on the detection of traffic agents within bounding boxes. They often overlook crucial environmental elements—such as lane markings, pedestrian paths, and traffic signs—that are not captured by rigid bounding box constraints, thereby failing to leverage a broader spectrum of visual information and contextual cues.

**Secondly**, there exists a propensity within numerous models to accord equal significance to all detectable entities within a traffic scene, an approach that might neglect the layered semantic interrelations that subsist among different entities. Such an approach risks overlooking essential insights that could significantly enhance the precision of accident anticipation

**Thirdly**, observational constraints intrinsic to real-world driving scenarios introduce substantial impediments. These encompass limitations of sensory apparatuses and environmental contingencies such as obstructions, adverse meteorological conditions, or traffic congestion. The majority of prevailing models, calibrated and assessed under conditions of optimal observational integrity, exhibit pronounced performance diminution when confronted with data-deficient scenarios. This discordance underscores a critical lacuna within contemporary research, necessitating models that manifest robust performance under suboptimal observational conditions.

In response to these articulated challenges, the present study introduces "CRASH," an avant-garde accident anticipation framework that meticulously integrates global contextual information with profound spatio-temporal interactions. This initiative is spearheaded by the introduction of a novel Object Focus Attention (OFA) mechanism within the object-aware module, which adeptly refines and accentuates key local features, extrapolating the essential spatial-temporal dynamics pivotal for accident prediction. Moreover, we pioneer a context-aware module that harnesses Fast Fourier Transform (FFT) along with our innovatively devised Context-aware Attention Blocks (CAB). This ensemble endeavors to distill nuanced global visual information, thereby amplifying the model's contextual comprehension and broadening the ambit of visual cues

amenable for predictive analysis. To sum up, our contributions are threefold:

(1) We present a novel context-aware module that extends global interactions into the frequency domain using FFT and introduces **context-aware attention blocks** to compute fine-grained correlations between nuanced spatial and appearance changes in different objections. Enhanced by the proposed **multi-layer fusion**, this framework dynamically prioritizes risks in various regions, enriching visual cues for accident anticipation.

(2) To realistically simulate the variability and randomness of missing data that is commonly encountered in real-world driving, we augment the renowned DAA, A3D, and CCD datasets with scenarios featuring **missing data**. This innovation expands the research scope for accident detection models and provides comprehensive benchmarks for evaluating model performance.

(3) In benchmark tests conducted on the enhanced DAD [5], A3D [39], and CCD [2] datasets, CRASH demonstrates superior performance over state-of-the-art (SOTA) baselines across key metrics, such as Average Precision (AP) and mean Time-To-Accident (mTTA). This showcases its remarkable accuracy and applicability across a variety of challenging scenarios, including those with **10%-50% data-missing** and **limited 50%-75% training set** scenes.

## 2 RELATED WORK

The task of predicting traffic accidents requires models capable of making timely and accurate predictions based on dashboard video before accidents occur. This task is made complex by the inherent variability of traffic scenes and the unpredictable movements of road users.

The surge in deep learning applications within computer vision has catalyzed the exploration of advanced models for accident anticipation. To tackle these challenges, recent studies have leveraged various deep learning approaches, including Convolutional Neural Networks (CNNs) [5, 10, 15, 21, 25], sequential networks [11, 31, 36, 37, 40? , 41] like Recurrent Neural Networks (RNNs), Long Short-Term Memory (LSTM) units, and Gated Recurrent Units (GRUs) to distill essential visual features from traffic scenes. Moreover, Graph Neural Networks (GNNs) [18, 23, 33, 34, 39] and transformer-based models [12, 35] have been investigated for their potential to capture the complex spatial and temporal dynamics in traffic scenes. In addition, generative models [3, 38] such as Generative Adversarial Networks (GANs), Variational Auto Encoders (VAEs), and Diffusion models are also employed in this field. For instance, Corcoran et al. [7] presented a dynamic-attention recurrent CNN to analyze both spatial and temporal features in traffic scenes. Similarly, Bao et al. [2] utilized an uncertainty-aware graph to model spatial relationships and predict traffic accidents, while Liu et al. [22] focused on pedestrian intent prediction through spatio-temporal analysis.

As research in traffic accident prediction deepens, a prominent challenge emerges: how to effectively manage and interpret the vast amount of information processed by models when dealing with complex traffic scenes. In this context, the incorporation of attention mechanisms [17, 18] marks a significant advance in the field, enhancing the ability of models to process complex interactions and maintain temporal coherence, thereby improving prediction

accuracy and interpretability. In particular, the efforts of Karim et al. [17] and Song et al. [29] have integrated spatial and temporal attention to prioritise relevant segments and regions in driving scenes. Thakare et al. [32] proposed a convolutional autoencoder approach for efficient feature extraction and classification, addressing computational efficiency. Additionally, the interpretability of models has gained prominence in research. Monjuru et al. [26] introduced an explainable artificial intelligence (XAI) strategy, embedding the Grad-CAM (Gradient-weighted Class Activation Mapping) attention mechanism within the GRUs to produce semantic feature maps.

Despite these advances, most studies focus on interactions between dynamic objects, overlooking crucial scene elements such as traffic lights, pedestrian crossings, and sidewalks. Furthermore, they typically rely on surface-level visual features that are close to the accidents, failing to adequately capture potential accident precursors in global scenes. Our work aims to fill this gap by integrating key scene elements and multi-layered features into our proposed model, thereby enriching the visual scope of accident detection. This integration allows for the capture of a wider range of semantic information, significantly improving the model's ability to anticipate traffic accidents with improved accuracy and timeliness.

## 3 METHODOLOGY

### 3.1 Problem Formulation

The primary objective of this study is twofold: (1) to predict the probability of a traffic accident occurring, and (2) if an accident does occur, to predict it as early as possible. Similar to the previous work [2], taking the $T$ frames of the dashboard video stream $V = \{V_1, V_2, \ldots, V_T\}$ as input, the goal is to estimate the probability $P = \{p_1, p_2, \ldots, p_T\}$ of an accident in each frame. If an accident occurs at time $t \in [1, T]$, we define the Time-to-Accident (TTA) as $\Delta t = \tau - t^o$, where $\tau$ is the ground-truth accident time, and $t^o$ is the earliest frame in which the probability score $p^t$ exceeds a predetermined threshold $p^o$. Consequently, a video is classified as containing an accident (positive) if $p^t \geq p^o$ for any $t \geq t^o$ and as not containing an accident (negative) if $\tau = 0$. Our proposed model aims to enhance the precision of accident detection and maximise the TTA, enabling the earliest possible anticipation of accidents.

### 3.2 Framework Overview

The overall pipeline of CRASH is shown in Fig. 1. It consists of five critical components: object detector, feature extractor, object-aware module, context-aware module, and multi-layer fusion. Initially, the object detector and feature extractor produce the object $F_o$ and context $F_c$ vectors for the raw input videos $V$. Next, the object-aware module is used to progressively update the spatial-temporal representation of the object vectors, producing the object-aware vectors $\bar{F}_o$. In parallel, the context vectors $F_c$ are fed into the context-aware modules for global semantic feature extraction, resulting in the context-aware vecotors $\bar{F}_c$. Finally, the multi-layer fusion iteratively fuses and mulls over the output from the feature extractor and these modules to identify and predict potential incidents that could lead to accidents, generating the probability $P$ for each frame of the input videos.

**Object Detector.** Given $T$-frames dashboard video, a Cascade R-CNN [4] is employed to detect the top-$n$ dynamic objects with the

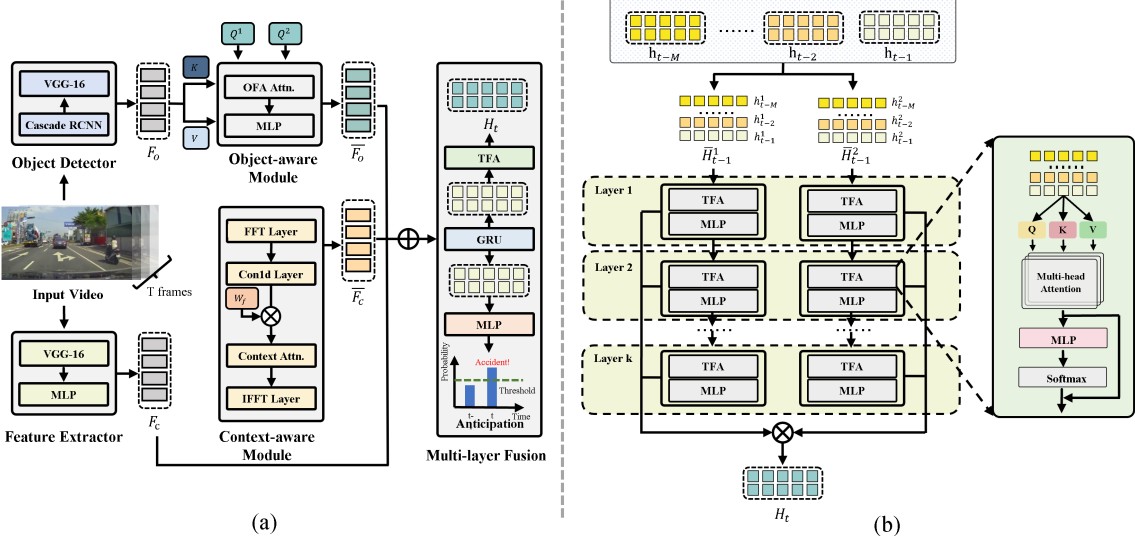

Figure 1: The overall framework of CRASH (a) and the architecture of Temporal Focus Attention (b).

highest recognition scores within the video stream, such as vehicles, motorcycles, and pedestrians. Then, we utilize the VGG-16 [28] to embed these selected $n$ objects into 2D object vectors $F_o \in \mathbb{R}^{n \times d}$, where $d$ is the embedding dimension.

**Feature Extractor.** The feature extractor is primarily responsible for extracting the semantic feature from the whole video $V$. The VGG-16 and Multilayer Perception (MLP) are used in this extractor to generate the context vectors $F_c \in \mathbb{R}^d$.

**Object-aware Module.** Accidents usually occur due to specific interactions between dynamic traffic agents, which are marked by decreasing spatial distance or irregular trajectories. Therefore, it is necessary to analyze each object's position and past movements, integrating data across both time and space. In this module, we leverage the object vectors $F_o$ and the weighted dual-layer hidden states $\bar{H}_{t-1} = \{\bar{H}_{t-1}^1, \bar{H}_{t-1}^2\} \in \mathbb{R}^{n \times d}$ encoded by the two-layer GRU and our proposed TFA attention mechanism in the multi-layer fusion to focus on the traffic agents most likely to cause accidents.

Specifically, we propose a query-centric Object Focus Attention (OFA) mechanism that maps dual-layer hidden states $\bar{H}_{t-1}^1, \bar{H}_{t-1}^2$ to distinct query values $Q_t^1, Q_t^2$ at time step $t-1$, each assigned unique linear projection weights. This process facilitates the calculation of spatial-temporal relationships between object vectors $F_o$ and their associated contextual and semantic features within the hidden states $H_{t-1}$. At time step $t$, this process can be defined as follows:

$$\begin{cases} Q_t^1 = W_Q^1(\phi_{MLP}(\bar{H}_{t-1}^1)) \\ Q_t^2 = W_Q^2(\phi_{MLP}(\bar{H}_{t-1}^2)) \\ K_t = W_K(\phi_{MLP}(F_o)) \\ V_t = W_V(\phi_{MLP}(F_o)) \end{cases} \tag{1}$$

where $W_Q^1 \in \mathbb{R}^{d \times d}$, $W_Q^2 \in \mathbb{R}^{d \times d}$, $W_K \in \mathbb{R}^{d \times d}$, and $W_V \in \mathbb{R}^{d \times d}$ are all learnable weights. $\phi_{MLP}$ denotes the MLP. Furthermore, We make matrix product on $K_t$ and $V_t$ and use the generated similarity query vectors $Q_t^1, Q_t^2$ to weight the object vectors $F_o$, producing the

enhanced object-aware vectors $\bar{F}_o$. Mathematically,

$$\bar{F}_o = \phi_{Softmax}\left(\frac{W_\alpha Q_t^1 K_t^{\mathrm{T}} + W_\beta Q_t^2 K_t^{\mathrm{T}}}{\sqrt{d_k}}\right) V_t \tag{2}$$

where $W_\alpha$ and $W_\beta$ are both the linear projection weights for the query vectors. Moreover, $\phi_{Softmax}$ represents the Softmax activation function, and $d_k$ is the projection channel dimension.

**Context-aware Module.** In addition to establishing spatio-temporal relationships between dynamic objects, modelling the visual context of detected objects - such as lane markings, sidewalks, and traffic signs - is crucial for distinguishing potential accident causes from others. To the end, we present a context-aware module that uses global context vectors to capture a wider range of visual cues and contextual features. This module not only identifies focal points within the input videos but also recognizes broader contextual relationships within the entire visual scene, going beyond the limitations of bounding boxes.

Departing from traditional methods that emphasize temporal features, this module focuses on spectral features. Inspired by the spectral and hierarchical transformers [14, 27], we first use the 1D convolutional layer to expand the number of channels to $c$ for the context vectors $F_c$. Thereafter, FFT is applied to transform context vectors into the Fourier domain, transitioning from temporal to spectral space. We then employ a parametrically learnable Spectral Gating Unit (SGU) alongside innovative context-aware attention blocks. This structure assigns weights to each frequency, enhancing the detection of subtle edge and contour details within the global visual scene. Formally,

$$S_c = W_f \cdot \phi_{FFT}(\phi_{conv1D}(F_c)) \tag{3}$$

Here, $\phi_{FFT}$ represents the Fast Fourier Transform (FFT) function, while $\phi_{conv1D}$ denotes a one-dimensional convolutional layer. Furthermore, $W_f \in \mathbb{R}^{c \times w \times h}$ is a learnable weight matrix produced

by the SGU. The spectral features, denoted as $S_c \in \mathbb{R}^{c \times h \times w}$, correspond to the context vectors $F_c$, where $c$ is the number of channels and $h$ and $w$ are the height and width of the feature map, respectively. Importantly, since the spectral features $S_c$ are complex numbers rather than real numbers, they cannot be directly subjected to gradient calculation and backpropagation. To address this, the Context-aware Attention Block and Inverse Fast Fourier Transform (IFFT) are introduced to further enhance the spectral features and transform the spectral space back to physical space. Furthermore, the features are processed by a MLP to eventually generate the vision-conditioned context-aware vectors $\bar{F}_c$, which improve the training stability of our model. It can be represented as follows:

$$
\begin{aligned}
\bar{F}_c &= \phi_{IFFT}(\phi_{CAB}(S_c)) \\
&= \phi_{IFFT}\left(\phi_{Softmax}(\phi_{MLP}\left[\phi_{AvgPool}(S_c) \oplus \phi_{MaxPool}(S_c)\right] \odot S_c)\right)
\end{aligned}
\tag{4}
$$

where $\phi_{IFFT}$ denotes the IFFT function, while $\oplus$ and $\odot$ signify the concatenation operation and element-wise multiplication, respectively. Correspondingly, $\phi_{AvgPool}$ and $\phi_{MaxPool}$ are the average and maximum pooling layers, respectively. In addition, the context-aware vectors $\bar{F}_v \in \mathbb{R}^d$, with embedding dimension $d = h \times w$.

**Multi-layer Fusion.** Accidents can occur unexpectedly at any moment in traffic scenes, and typically occupy a small proportion of the entire video stream, exhibiting a long-tail distribution. Most existing methods tend to focus on anomalies within key frames of the video stream. They directly feed the top-layer frame features into a linear layer to predict the probability of an accident. However, certain frames that are close to anomalous moments, despite lacking direct abnormal phenomena, often contain enriched contextual information that is crucial for assessing the likelihood of an accident. To fully exploit the semantic and contextual features embedded in every frame of the input video, we introduce the Temporal Focus Attention (TFA) mechanism. Comprising $k$ attention layers, each with two attention blocks as depicted in Fig. 1, this mechanism systematically integrates representations from diverse frame before preceding the prediction of accident probabilities. The core idea is to expand the model's recognition scope and reference range to all visual information in the video stream, focusing dynamically on the embedded features at each moment to improve the model's ability to identify key frames in input video.

From a technical perspective, the context $F_c$, object-aware $\bar{F}_o$, and context-aware $\bar{F}_c$ vectors are fused and encoded by a dual-layer GRU, which can be expressed as follows:

$$
O_t, \; h_t^i = \phi_{GRU}(F_c \| \bar{F}_c \| \bar{F}_o)
\tag{5}
$$

where $\|$ signifies the vector concatenation, while $h_t^i$ represents the hidden state of the $i$-th layer GRU at time step $t$, and $O_t^i$ is the GRU's final output for the $t$-th frame video, subsequently input into a MLP to generate the accident probability score $p_t$.

In the TFA layer, each block inputs the hidden states produced by the dual-layer GRU from the past $M$ frames, denoted as $H_t = \{H_t^1, H_t^2\}$, with each $H_t^i = \{h_{t-M}^i, \ldots, h_{t-2}^i, h_{t-1}^i\} \in \mathbb{R}^{M \times d}$, $i \in [1, 2]$. Furthermore, these hidden states are projected into query $\bar{Q}_t^i$, key $\bar{K}_t^i$, value $\bar{V}_t^i$ vectors. Formally,

$$
\bar{Q}_t^i = \bar{W}_Q^i H_t^i, \quad \bar{K}_t^i = \bar{W}_K^i H_t^i, \quad \bar{V}_t^i = \bar{W}_V^i H_t^i
\tag{6}
$$

where $\bar{W}_Q^i, \bar{W}_K^i, \bar{W}_V^i \in \mathbb{R}^{d \times d}$ are learnable matrices for the linear projection. The $j$-th attention head $head_j^i$ and the output $R_y^i$ from $y$-th RTA layer's attention block is computed as:

$$
R_y^i = \sum_{j=1}^{m} head_j^i = \sum_{j=1}^{m} \phi_{Softmax}\left(\frac{\bar{Q}_t^i (\bar{K}_t^i)^T}{\sqrt{\bar{d}_k}}\right) \odot \bar{V}_t^i
\tag{7}
$$

where $m$ is the total number of the attention head. Inspired by ResNet[16], the TFA block integrates Gated Linear Units (GLUs) [8] for optimizing the output. This ensures effective backpropagation of larger gradients to the initial layers, facilitating these layers to learn as rapidly as the top layer. Mathematically,

$$
\bar{R}_y^i = \phi_{Softmax}(\phi_{MLP}(\phi_{GLUs}(R_y^i))) + R_y^i
\tag{8}
$$

where $\bar{R}_y^i \in \mathbb{R}^{m \times d}$ is the enhanced output of $y$-the TFA layer for $i$-th attention block, and $\phi_{GLUs}$ is the GLUs function.

Finally, the outputs of $i$-th attention blocks $\bar{R}_1^i, \bar{R}_2^i, \ldots, \bar{R}_k^i$ across $k$ attention layers are dynamically aggregated using distinct learnable weights to obtain the final hidden state $\bar{H}_t$. This process is formalized as follows:

$$
\bar{R}_{weighted}^i = \gamma_1^i \cdot R_1^i + \gamma_2^i \cdot R_2^i + \ldots + \gamma_k^i \cdot R_k^i
\tag{9}
$$

where $\gamma_1^i$, $\gamma_2^i$, and $\gamma_k^i, i \in [1, 2]$ are the learnable parameters. The final hidden states of the TFA layer can be defined mathematically as $H_t = \phi_{AvgPool}(\bar{R}_{weighted}^1) \oplus \phi_{AvgPool}(\bar{R}_{weighted}^2)$, which are fed into the Object Focus Attention (OFA) in the object-aware module for feature fusion.

## 3.3 Training Loss

We incorporate a multi-task learning paradigm into our training loss, which can be bifurcated into two components: (1) *anticipation loss* $\mathcal{L}_a$ and (2) *enhancement loss* $\mathcal{L}_e$.

The *anticipation loss* $\mathcal{L}_a$ is computed based on the discrepancy between the model-predicted accident probabilities $p_t$ at time step $t$ and the ground-truth accident timing $\tau$. To better align with the task of real-world traffic accident anticipation, we refine the traditional cross-entropy loss function in *anticipation loss* by integrating a penalty term $e^{-\frac{1}{2}\max(\frac{\tau-t}{f}, 0)}$ into the positive loss component. This adjustment applies increasing loss values to video stream that are closer to the moment of an accident, encouraging the model to predict accidents earlier. The *anticipation loss* $\mathcal{L}_a$ is expressed as follows:

$$
\mathcal{L}_a = \frac{1}{B} \sum_{v=1}^{B} \left[ -l_v \sum_{t=1}^{T} e^{-\frac{1}{2}\max(\frac{\tau-t}{f}, 0)} log(p^t) - (1 - l_v) \sum_{t-1}^{T} log(1 - p^t) \right]
\tag{10}
$$

where $B$ is the batch size, $l_v$ represents the binary label of accident occurrence within each video (1 for an accident, 0 for none), while $T$ is the total number of frames per video, and $f$ is the frames per second (fps) of the video.

Furthermore, we introduce an innovative *enhancement loss*, $\mathcal{L}_e$, to mitigate the significant error accumulation in the initial stages of the GRU within the multi-layer fusion. Specifically, position encoding is integrated into all hidden states produced by the second-layer GRU, and a classical multi-head self-attention mechanism is employed to extract relevant semantic information from these hidden states, resulting in the hidden state maps $p_e$:

$$
p_e = \phi_{MLP}(\phi_{MHA}(\phi_{PE}(h_1^2, h_2^2, \ldots, h_T^2)))
\tag{11}
$$

where $\phi_{MHA}$ and $\phi_{PE}$ denote the multi-head attention mechanism and position encoding mechanism, respectively.

Next, we compute the *enhancement loss* $\mathcal{L}_e$ using the hidden state maps $p_e$ and the ground-truth accident timing $\tau$. Formally,

$$\mathcal{L}_e = \frac{1}{B} \sum_{v=1}^{B} \left[ -l_v \log(p_e) - (1 - l_v) \log(1 - p_e) \right] \quad (12)$$

Eventually, the final loss is then calculated as the sum of *anticipation loss* $\mathcal{L}_a$ and *enhancement loss* $\mathcal{L}_e$, adjusted for homoscedastic uncertainty through Gaussian probability:

$$\mathcal{L} = \frac{\mu_1}{2\rho_1^2} \mathcal{L}_a + \frac{\mu_2}{2\rho_2^2} \mathcal{L}_e + \log(\rho_1 \rho_2) \quad (13)$$

where $\mu_1$ and $\mu_2$ are manually-set hyperparameters, while $\rho_1$ and $\rho_2$ represent uncertainty coefficients, initially set to 1. Overall, the multi-task training loss is meticulously designed to provide a dynamic balance between anticipating accidents and enhancing model sensitivity to critical features, thus taking into account more accident-related factors.

## 4 EXPERIMENT

### 4.1 Experiment Setup

We evaluate the efficacy of our model using three esteemed datasets: Dashcam Accident Dataset (DAD), Car Crash Dataset (CCD), and AnAn Accident Detection (A3D) datasets. These datasets, referred to as *complete* datasets, provide a unique perspective on traffic accidents in various scenes. More details can be found in the **Appendix**.

Recognizing a gap in research concerning data omissions in this field, the experiment setup is intentionally designed to simulate the variability and randomness of missing data encountered in real-world scenarios. Specifically, we propose three specialized versions of each primary dataset, referred to as *missing* datasets: DAD-missing, A3D-missing, and CCD-missing. These datasets are meticulously crafted to realistically mimic the variability and randomness of data omissions encountered in real-world settings. They include emulated missing observation rates of 10%, 20%, and 50%, as well as a fixed pattern of missing one or two frames every five frames (1/2 in 5 frames). These scenarios cover a broad spectrum of potential data loss situations, from minimal to severe. A stochastic mechanism is used to determine which observations are missing, avoiding the introduction of bias and more accurately reflecting the unpredictability inherent in real-world data collection. To evaluate the adaptability and effectiveness of our model, we conduct training on reduced versions of the datasets, specifically 50% and 75% subsets. We then evaluate the performance of our proposed model on both *complete* and *missing* datasets. These evaluations aim to gauge the model's adaptability to unfamiliar data and its proficiency in handling data omissions, providing a comprehensive evaluation of the robustness of our proposed model.

### 4.2 Evaluation Metrics

This study evaluates model performance by considering both the accuracy (Average Precision) and timeliness (Time-to-Accident) of model predictions.

**Accuracy**. Accident detection accuracy of the model is quantified by recall (R), which is defined as the ratio of correctly identified

**Table 1: Comparison of models seeking balance between mTTA and AP on the *complete* datasets. Bold and underlined values represent the best and second-best performance in each category. Instances where values are not available are marked with a dash ("-").**

| Model | DAD [5] | | CCD [2] | | A3D [39] | |
|---|---|---|---|---|---|---|
| | AP(%)↑ | mTTA(s)↑ | AP(%)↑ | mTTA(s)↑ | AP(%)↑ | mTTA(s)↑ |
| DSA [5] | 48.1 | 1.34 | **99.6** | 4.53 | 93.4 | 4.41 |
| L-RAI [42] | 51.4 | 3.01 | 98.9 | 3.32 | - | - |
| AdaLEA [30] | 52.3 | 3.43 | 99.2 | 3.45 | 92.9 | 3.16 |
| DSTA [17] | 59.2 | 2.60 | **99.6** | 4.87 | 94.2 | 4.81 |
| UString [2] | 53.7 | **3.53** | 99.5 | 4.73 | 94.4 | **4.92** |
| GSC [34] | 60.4 | 2.55 | 99.3 | 3.58 | 94.9 | 2.62 |
| **CRASH** | **65.3** | 3.05 | **99.6** | 4.91 | **96.0** | **4.92** |

accident videos (true positives, TP) to the actual number of accident videos (TP plus false negatives, FN). Prediction reliability is assessed by precision (P), the ratio of TP to the sum of TP and false positives (FP). To account for how recall and precision fluctuate with threshold adjustments, we use average precision (AP) as an overall measure of model accuracy. It calculated as the area under the precision-recall curve $AP = \int P(R) \, dR$, serves as an overall indicator of the model's consistency in making accurate predictions across different threshold levels, with higher AP values indicating superior performance.

**Timeliness.** The Time-to-Accident (TTA) is the metric used to evaluate the model's predictive timeliness. It measures the interval between the model's initial accident prediction (once the risk level surpasses a pre-set threshold) and the actual occurrence of the accident. A greater TTA indicates that the model can foresee accidents well in advance, providing drivers with more response time. The Mean Time-to-Accident (mTTA) calculates the average TTA values across various thresholds. Under strict recall rate conditions, we also evaluate the model's early warning effectiveness at a recall of 80%, referred to as TTA@R80.

### 4.3 Implementation Details

The proposed model is implemented using PyTorch and trained on an NVIDIA A40 (48GB) GPU over 80 epochs with a consistent batch size of 10. We use the Adam optimiser, initialising the learning rate at $1 \times 10^{-4}$ uniformly across all datasets. The object detector is configured to detect up to 19 candidate objects, and the embedding dimension for VGG-16 is set to 4096, and the hidden state dimension of the GRU is fixed at 512. In addition, the ReduceLROnPlateau strategy is used to schedule the learning rate, which adjusts the rate in response to the model's performance across epochs. See **Appendix** for more implementation details.

### 4.4 Evaluation Results

**Compare with SOTA Baselines on *Complete* Datasets.** Table 1 illustrates that our model exhibits SOTA performance across all metrics on the DAD, A3D, and CCD datasets for considering the trade-off between timeliness (mTTA) and accuracy (AP) of accident anticipation. Specifically, on the CCD and A3D datasets, our model's AP and mTTA metrics have already reached or exceeded all baselines. On the DAD dataset, which covers a wide range of

**Table 2: Comparison of models for the evaluation metrics on *missing* datasets. @R80 refers to the TTA@R80, which represents the value of mTTA at a recall of 80%. Bold and underlined values represent the best and second-best performance.**

| Dataset | Model | drop-10% | | | drop-20% | | | drop-50% | | | 1 in 5 | | | 2 in 5 | | |
|---|---|---|---|---|---|---|---|---|---|---|---|---|---|---|---|---|
| | | AP(%)↑ | mTTA(s)↑ | @R80(s)↑ | AP(%)↑ | mTTA(s)↑ | @R80(s)↑ | AP(%)↑ | mTTA(s)↑ | @R80(s)↑ | AP(%)↑ | mTTA(s)↑ | @R80(s)↑ | AP(%)↑ | mTTA(s)↑ | @R80(s)↑ |
| DAD [5] | Ustring [2] | 53.51 | 2.50 | 2.81 | 52.27 | 2.47 | 2.24 | 52.37 | 1.62 | 2.26 | 52.62 | 2.14 | 1.73 | 48.86 | 1.86 | 1.77 |
| | DSTA [17] | 56.78 | 2.48 | 2.90 | 55.89 | 2.48 | 2.90 | 54.84 | 2.11 | 2.89 | 55.46 | 2.59 | 2.76 | 53.01 | 2.16 | 3.05 |
| | GSC [34] | 55.21 | 2.56 | 2.46 | 54.78 | 2.35 | 2.62 | 51.39 | 1.81 | 2.64 | 55.59 | 2.21 | 2.91 | 50.87 | 2.15 | 2.57 |
| | **CRASH** | 65.24 | 2.84 | 3.13 | 64.64 | 2.76 | 2.99 | 63.34 | 2.37 | 2.94 | 64.38 | 2.51 | 3.09 | 64.39 | 2.40 | 3.05 |
| A3D [39] | UString [2] | 94.01 | 4.74 | 4.21 | 93.11 | 4.59 | 4.10 | 91.26 | 3.66 | 3.18 | 93.48 | 4.34 | 3.41 | 92.62 | 3.81 | 3.23 |
| | DSTA [17] | 93.77 | 4.82 | 4.30 | 92.31 | 4.82 | 4.13 | 91.80 | 3.75 | 3.60 | 93.54 | 4.57 | 3.63 | 91.33 | 3.70 | 3.45 |
| | **CRASH** | 95.96 | 4.88 | 4.81 | 94.83 | 4.77 | 4.20 | 94.54 | 4.24 | 4.18 | 94.88 | 4.74 | 4.56 | 95.41 | 4.81 | 4.58 |
| CCD [2] | UString [2] | 98.71 | 4.73 | 4.22 | 96.44 | 4.36 | 3.58 | 94.52 | 4.39 | 3.81 | 96.79 | 4.44 | 3.79 | 94.82 | 4.60 | 4.17 |
| | DSTA [17] | 98.80 | 4.79 | 4.31 | 97.18 | 4.51 | 3.82 | 94.73 | 4.01 | 3.02 | 97.95 | 4.53 | 3.83 | 96.18 | 4.32 | 3.57 |
| | **CRASH** | 99.30 | 4.89 | 4.54 | 98.93 | 4.69 | 4.50 | 98.46 | 4.53 | 4.28 | 98.91 | 4.76 | 4.42 | 98.78 | 4.61 | 4.11 |

**Table 3: Comparison of models trained with limited training sets on evaluation metrics for *missing* dataset.**

| Dataset | Model | Drop-10% | | | Drop-20% | | | Drop-50% | | | 1 in 5 Frames | | | 2 in 5 Frames | | |
|---|---|---|---|---|---|---|---|---|---|---|---|---|---|---|---|---|
| | | AP(%)↑ | mTTA(s)↑ | @R80(s)↑ | AP(%)↑ | mTTA(s)↑ | @R80(s)↑ | AP(%)↑ | mTTA(s)↑ | @R80(s)↑ | AP(%)↑ | mTTA(s)↑ | @R80(s)↑ | AP(%)↑ | mTTA(s)↑ | @R80(s)↑ |
| DAD [5] (75%) | Ustring [2] | 55.10 | 2.61 | 3.20 | 49.52 | 2.45 | 2.65 | 45.94 | 2.24 | 2.67 | 47.57 | 2.85 | 3.99 | 45.48 | 2.46 | 3.11 |
| | DSTA [17] | 54.12 | 2.40 | 2.91 | 53.13 | 2.59 | 3.09 | 50.52 | 2.16 | 2.91 | 53.24 | 2.55 | 3.16 | 49.43 | 2.48 | 2.76 |
| | GSC [34] | 54.37 | 2.57 | 3.22 | 51.51 | 2.35 | 3.10 | 49.18 | 2.21 | 2.58 | 50.54 | 2.24 | 3.05 | 49.72 | 2.28 | 3.07 |
| | **CRASH** | 62.46 | 2.64 | 3.31 | 60.04 | 2.57 | 3.32 | 58.37 | 2.31 | 3.02 | 61.25 | 2.66 | 3.27 | 57.91 | 2.56 | 3.18 |
| A3D [39] (75%) | UString [2] | 94.10 | 4.21 | 4.28 | 93.90 | 3.80 | 4.24 | 92.58 | 3.09 | 3.49 | 93.84 | 3.90 | 3.81 | 91.75 | 4.32 | 4.21 |
| | DSTA [17] | 91.37 | 4.32 | 3.38 | 91.15 | 4.19 | 4.05 | 89.70 | 3.52 | 3.84 | 91.36 | 3.90 | 4.77 | 90.25 | 3.77 | 3.25 |
| | **CRASH** | 95.61 | 4.82 | 4.60 | 95.42 | 4.71 | 4.47 | 93.63 | 4.41 | 3.98 | 94.48 | 4.65 | 4.01 | 94.14 | 4.65 | 4.38 |
| CCD [2] (75%) | Ustring [2] | 96.63 | 4.68 | 4.17 | 95.23 | 4.48 | 3.79 | 94.43 | 4.15 | 3.57 | 94.24 | 4.22 | 4.27 | 93.31 | 3.75 | 3.07 |
| | DSTA [17] | 97.94 | 4.24 | 4.44 | 95.85 | 4.49 | 3.28 | 95.37 | 3.91 | 3.20 | 96.61 | 4.02 | 2.57 | 94.66 | 4.14 | 3.30 |
| | CRASH | 98.13 | 4.72 | 4.44 | 97.37 | 4.65 | 4.32 | 96.90 | 4.23 | 3.88 | 97.00 | 4.36 | 4.41 | 95.94 | 4.29 | 3.67 |
| DAD [5] (50%) | Ustring [2] | 53.22 | 2.36 | 2.70 | 52.05 | 2.51 | 3.96 | 50.39 | 2.24 | 2.79 | 50.80 | 2.43 | 2.89 | 48.68 | 2.22 | 2.27 |
| | DSTA [17] | 51.64 | 2.62 | 2.26 | 49.97 | 2.24 | 2.67 | 46.03 | 1.90 | 2.46 | 51.19 | 1.89 | 2.96 | 46.45 | 2.13 | 2.70 |
| | GSC [34] | 54.18 | 2.59 | 2.79 | 52.98 | 2.39 | 3.30 | 51.09 | 1.94 | 2.64 | 51.39 | 2.06 | 2.88 | 50.43 | 2.15 | 2.66 |
| | **CRASH** | 58.22 | 2.70 | 3.01 | 57.60 | 2.58 | 3.31 | 57.71 | 2.28 | 3.20 | 58.73 | 2.32 | 3.07 | 58.11 | 2.26 | 3.13 |
| A3D [39] (50%) | UString [2] | 92.23 | 4.48 | 3.96 | 92.25 | 4.47 | 4.11 | 91.59 | 3.98 | 3.99 | 91.75 | 4.31 | 4.18 | 90.29 | 4.28 | 4.17 |
| | DSTA [17] | 89.26 | 4.08 | 4.24 | 88.88 | 4.03 | 3.86 | 86.70 | 3.76 | 3.20 | 90.48 | 4.05 | 3.71 | 87.12 | 3.60 | 4.02 |
| | **CRASH** | 94.98 | 4.80 | 4.43 | 93.67 | 4.59 | 3.97 | 92.47 | 4.63 | 4.59 | 94.32 | 4.75 | 4.49 | 93.87 | 4.46 | 4.77 |
| CCD [2] (50%) | Ustring [2] | 94.51 | 4.37 | 3.87 | 92.91 | 4.32 | 3.68 | 91.13 | 4.04 | 3.84 | 91.38 | 4.45 | 3.91 | 90.74 | 4.21 | 4.03 |
| | DSTA [17] | 96.68 | 4.21 | 2.65 | 95.79 | 4.19 | 4.40 | 95.52 | 3.92 | 2.47 | 96.09 | 4.20 | 3.56 | 94.65 | 4.27 | 3.13 |
| | **CRASH** | 97.31 | 4.37 | 4.12 | 97.08 | 4.46 | 4.08 | 96.78 | 4.31 | 3.95 | 97.08 | 4.46 | 3.95 | 96.07 | 4.35 | 4.16 |

**Table 4: Comparison of models for the best AP on DAD datasets. @R80 denotes the TTA@R80. Instances where values are not available are marked with a dash ("-").**

| Model | Backbone | Publication | AP(%)↑ | mTTA(s)↑ | @R80(s)↑ |
|---|---|---|---|---|---|
| L-RAI[42] | VGG-16 | ACCV'16 | 51.40 | - | - |
| DSA [5] | VGG-16 | ACCV'16 | 63.50 | 1.67 | 1.85 |
| UniFormerv2 [20] | Transformer | ICCV'23 | 65.24 | - | - |
| VideoSwin [24] | Transformer | CVPR'22 | 65.45 | - | - |
| MViTv2 [9] | Transformer | CVPR'21 | 65.45 | - | - |
| DSTA [17] | VGG-16 | TITS'22 | 66.70 | 1.52 | 2.39 |
| UString [2] | VGG-16 | ACM MM'20 | 68.40 | 1.63 | 2.18 |
| GSC [34] | VGG-16 | IEEE TIV'23 | 68.90 | 1.33 | 2.14 |
| **CRASH** | VGG-16 | - | 70.86 | 1.91 | 2.20 |

**Table 5: Comparison of models for the evaluation metrics on limited training sets. @R80 represents the TTA@R80.**

| Dataset | Model | 75% Training Set | | | 50% Training Set | | |
|---|---|---|---|---|---|---|---|
| | | AP(%)↑ | mTTA(s)↑ | @R80(s)↑ | AP(%)↑ | mTTA(s)↑ | @R80(s)↑ |
| DAD [5] | Ustring [2] | 51.15 | 2.67 | 2.40 | 52.68 | 2.38 | 2.74 |
| | DSTA [17] | 52.79 | 2.71 | 2.65 | 52.78 | 2.52 | 2.83 |
| | GSC [34] | 58.14 | 2.76 | 2.84 | 56.32 | 2.38 | 3.05 |
| | **CRASH** | 64.10 | 2.76 | 3.12 | 61.41 | 2.62 | 3.23 |
| A3D [39] | Ustring [2] | 94.28 | 4.54 | 3.69 | 93.30 | 4.58 | 4.01 |
| | DSTA [17] | 92.86 | 4.58 | 3.16 | 91.75 | 4.11 | 3.98 |
| | **CRASH** | 94.92 | 4.73 | 4.57 | 94.98 | 4.65 | 4.49 |
| CCD [2] | Ustring [2] | 98.55 | 4.77 | 4.28 | 97.05 | 4.27 | 4.36 |
| | DSTA [17] | 98.69 | 4.58 | 3.92 | 97.21 | 4.49 | 4.39 |
| | **CRASH** | 99.17 | 4.87 | 4.76 | 98.19 | 4.71 | 4.40 |

urban scenarios and traffic complexities, our model achieves an optimal AP of 65.30%, surpassing the second-ranked GSC model by 8.11%. Additionally, it maintained a competitive mTTA of 3.05 seconds. These results demonstrate our model's superior capability to navigate through complex and variable traffic scenes, including different levels of congestion, urban roads, and traffic conditions.

Furthermore, Table 4 presents a detailed comparison of our model against the top baselines on the DAD dataset, highlighting the model's superior performance. Our model achieves the highest AP value and the corresponding highest mTTA value within the 5-second accident detection horizon. This indicates an average lead time before an accident of 1.91 seconds, which is 14.37% higher than the second-place DSA, providing more time to take preventive measures. Notably, comparing mTTA values without considering AP is not practically significant in real-world driving scenarios. Therefore, our experiments exclude such comparisons.

**Performance Comparison on *Missing* Datasets.** Table 2 demonstrates the robustness of our model in handling missing observations. Our model significantly outperforms all other baselines when tested on datasets with 10%, 20% and 50% randomly missing datasets. On the A3D-missing, CCD-missing, and DDA-missing datasets, our model outperforms the leading models with an average improvement of at least 10.59% in AP and 6.69% in mTTA. With 10% of the data missing, our model outperforms all SOTA baselines tested on

complete data, demonstrating significantly higher values in both AP and mTTA metrics—evidencing its superior predictive capability.

As expected, the performance of the model is directly influenced by the amount of input data available. The more input frames that are omitted, the greater the impact on predictive performance. However, even in datasets with significant data omission (Drop-50%), our model's performance remained superior to other baselines and competitive with models tested on complete data. Furthermore, in datasets with continuous data loss (1 in 5 and 2 in 5 frames), our model's metrics were still better than most state-of-the-art (SOTA) baselines, demonstrating its robustness and broad applicability in real-world driving scenarios.

**Performance Comparison on Limited Training Sets.** To demonstrate the scalability and efficiency of our model, we train it and some open-source baselines on a reduced portion of the training sets (50% and 75%) and evaluate them on both complete and missing datasets. Our model significantly outperforms all other baselines, as detailed in Table 3, despite severe performance drops observed in top models like GSC and Ustring. Remarkably, our model still stands out with significantly higher AP and mTTA values across the board in *missing* data scenarios, even when trained on substantially less data, as shown in Table 5. This finding emphasizes the model's capacity to minimize training data requirements, showcasing its adaptability in situations characterized by data loss and fragmentation errors common in the perception process.

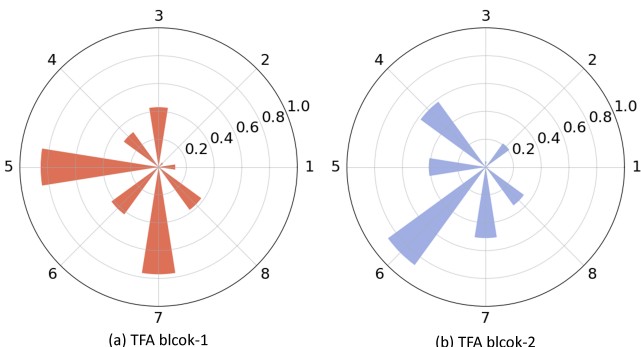

(a) TFA blcok-1    (b) TFA blcok-2

**Figure 2: Attention weights of hidden states over all TFA blocks in 8 TFA layers.**

## 4.5 Ablation Studies

**Ablation Study for Core Components.** Table 6 reports the ablation results of five critical components in CRASH: Object Focus Attention, Context-aware Attention Block, Fast Fourier Transform, Temporal Focus Attention, and enhancement loss $\mathcal{L}_e$.

Evaluation across the DAD, A3D, and CCD datasets shows that models lacking any of these components exhibit reduced performance, as evidenced by significant decreases in AP, mTTA, and TTA@80% metrics compared to the holistic model. In particular, the integration of CAB and FFT emerges as a significant performance enhancer, underscoring their indispensable role in the context-aware module to improve the model's ability to capture contextual

**Table 6: Ablation results for core components.**

| Dataset | OFA | CAB | FFT | TFA | $\mathcal{L}_e$ | AP(%)↑ | mTTA(s)↑ | @R80(s)↑ |
|---|---|---|---|---|---|---|---|---|
| DAD [5] | ✓ | ✓ | ✓ | ✓ | ✓ | 65.3 | 3.05 | 3.18 |
| | ✗ | ✓ | ✓ | ✓ | ✓ | 61.2 | 2.46 | 2.88 |
| | ✓ | ✗ | ✓ | ✓ | ✓ | 59.5 | 2.02 | 2.48 |
| | ✓ | ✓ | ✗ | ✓ | ✓ | 60.5 | 2.28 | 2.61 |
| | ✓ | ✓ | ✓ | ✗ | ✓ | 62.8 | 2.51 | 2.96 |
| | ✓ | ✓ | ✓ | ✓ | ✗ | 64.9 | 2.65 | 2.82 |
| A3D [39] | ✓ | ✓ | ✓ | ✓ | ✓ | 96.0 | 4.92 | 4.95 |
| | ✗ | ✓ | ✓ | ✓ | ✓ | 92.6 | 4.49 | 4.71 |
| | ✓ | ✗ | ✓ | ✓ | ✓ | 91.1 | 4.50 | 3.73 |
| | ✓ | ✓ | ✗ | ✓ | ✓ | 92.4 | 4.58 | 4.06 |
| | ✓ | ✓ | ✓ | ✗ | ✓ | 92.8 | 4.28 | 3.90 |
| | ✓ | ✓ | ✓ | ✓ | ✗ | 94.4 | 4.85 | 4.23 |
| CCD [2] | ✓ | ✓ | ✓ | ✓ | ✓ | 99.5 | 4.91 | 4.97 |
| | ✗ | ✓ | ✓ | ✓ | ✓ | 96.8 | 4.67 | 4.20 |
| | ✓ | ✗ | ✓ | ✓ | ✓ | 94.5 | 4.76 | 4.26 |
| | ✓ | ✓ | ✗ | ✓ | ✓ | 96.2 | 4.77 | 4.46 |
| | ✓ | ✓ | ✓ | ✗ | ✓ | 97.6 | 4.77 | 4.39 |
| | ✓ | ✓ | ✓ | ✓ | ✗ | 98.5 | 4.88 | 4.62 |

cues and thus improve prediction accuracy. Importantly, OFA significantly improves model performance by detecting critical interactions between traffic entities, which are essential for accurate accident prediction. It identifies potential anomalous intentions and movement trends of traffic agents within scenes, enabling the model to focus on the high-risk objectors. In addition, the inclusion of the enhancement loss $\mathcal{L}_e$ refines the hidden states of the model, while TFA enhances the semantic features within these states. This focus on key hidden states for anticipatory analysis, combined with the sophisticated information processing of other components, greatly improves the model's prediction reliability.

**Case Study for TFA Mechanism.** To further demonstrate the effectiveness of the proposed TFA mechanism, we visualize the distribution of attention weights across eight attention layers within two distinct TFA blocks. As illustrated in Fig. 2, the higher attention layers, particularly layers 4 to 8, receive a greater proportion of attention weights. This observation suggests that the influence on these upper layers increases with the number of attention layers, likely due to enhanced vector interactions. Contrary to initial expectations, the top layer does not dominate in terms of attention weights. Instead, the mid-upper layers (4-7) receive heightened attention, suggesting they may contain critical semantic features for accident prediction. This deviates markedly from traditional techniques that typically rely on the top layer's representations for accident prediction, which may overlook critical semantic features inherent in other layers.

In light of these findings, we introduced the TFA mechanism to allocate weights dynamically across different attention layers. This allocation is meticulously calibrated based on the continuous evolution of hidden states within the video sequence, ensuring that each layer contributes optimally based on its informational content. Subsequently, the multi-layer fusion iteratively fuses and updates these outputs from these diversified TFA attention layers using the assigned weights, providing the object-aware module with a wide range of visual cues. In summary, the TFA mechanism promotes a more nuanced understanding and exploitation of the hierarchical features within the network, enabling the detection of subtle cues that are critical for early and accurate accident anticipation.

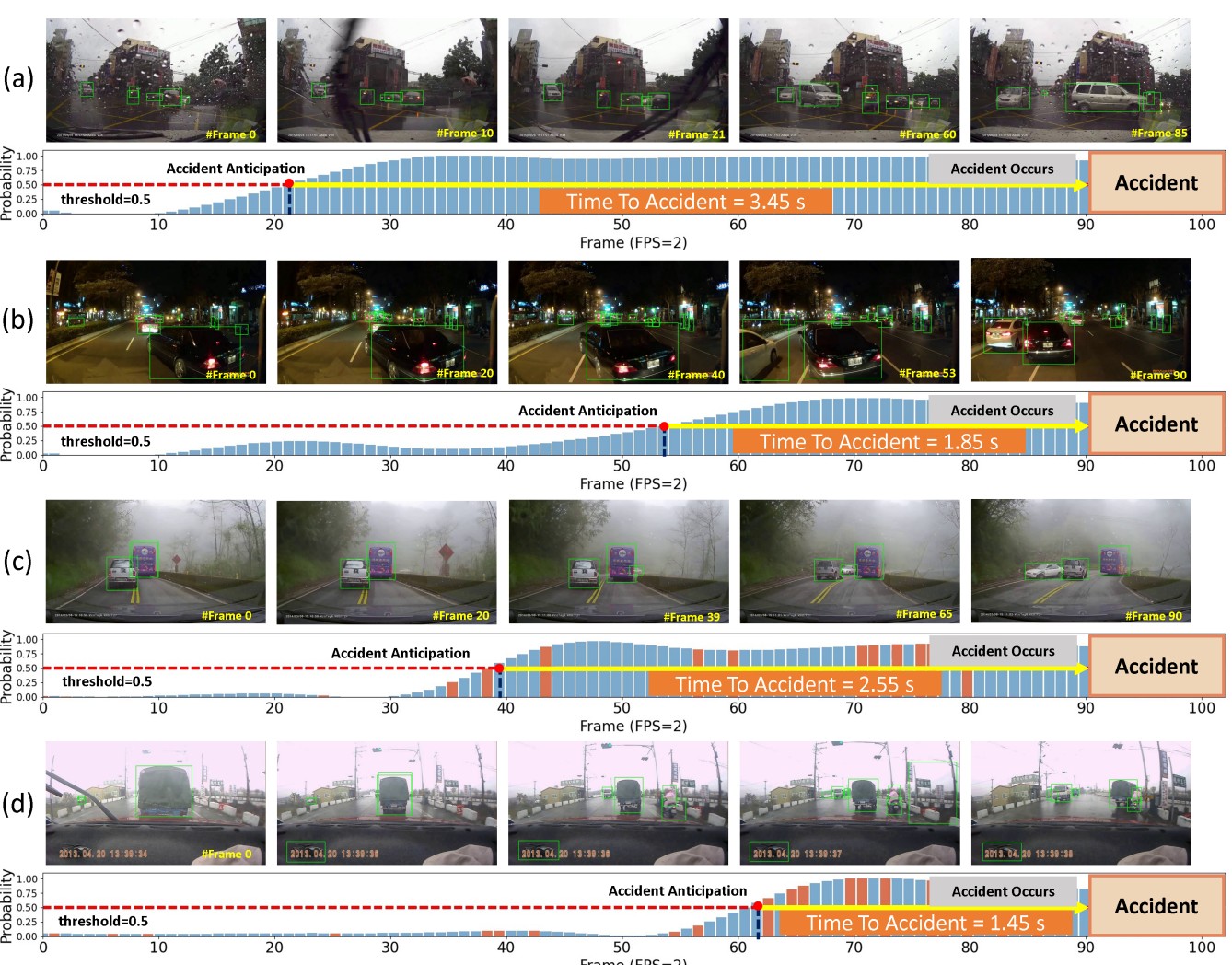

Figure 3: Qualitative Results of CRASH in rainy weather (a) and low nighttime lighting (b), heavy fog (c), and dense multi-agent traffic scenes (d) on the DAD dataset. The orange bar graph represents the loss of video data for that frame.

## 4.6 Qualitative Results

Fig. 3 illustrates the accident anticipation capabilities of our model in challenging real-world driving scenarios. CRASH demonstrates a consistent ability to accurately identify impending accidents across a wide range of environmental conditions and to issue timely warnings at least 3 seconds in advance of potential incidents (TTA>3) in *complete* datasets, as shown in Fig. 3 (a-b). Remarkably, even in scenarios featuring by data missing, as highlighted in Fig. 3 (c), our model calculates the likelihood of an accident in real-time with remarkable accuracy. In addition, Fig. 3 (d) reveals that our model remains capable of predicting accidents at least 1.45 seconds in advance (TTA>1.45) in scenarios with up to 20% missing data despite being trained on only 50% of the training set. These qualitative results highlight the exceptional robustness of the model and its potential to tackle corner-case traffic scenarios.

## 5 CONCLUSION

This study introduces a novel accident anticipation framework CRASH for autonomous driving, which integrates five key components: object detector, feature extractor, object-aware module, context-aware module, and multi-layer fusion. These components work in concert to analyze and interpret various features, capture the fine-grained correlations between different traffic agents, and account for the inherent uncertainty and long-tail effects in the accident anticipation. Rigorous evaluations conducted on the DAD, A3D, and CCD demonstrate the robustness and adaptability of CRASH, demonstrating its superior performance even in scenarios with data constraints and missing data. In addition, we introduce the enhancing versions of these datasets—DAD-missing, A3D-missing, and CCD-missing—to simulate the variability and randomness of real-world data omissions, further refining accident anticipation methodologies in data-missing scenes.

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
