# OpenReview forum: "CRASH: Crash Recognition and Anticipation System Harnessing with Context-Aware and Temporal Focus Attentions"
_acmmm.org/ACMMM/2024/Conference — MM2024 Poster_

### Official Review · Reviewer_PPcW · 2024-05-15

**Rating:** 3
**Confidence:** 3

**Summary:**

The paper presents CRASH, a novel accident anticipation framework designed for autonomous vehicles (AVs). It aims to predict accidents from camera footage by integrating an object detector, feature extractor, object-aware module, context-aware module, and a multi-layer fusion approach. The object-aware module focuses on high-risk objects, while the context-aware module uses Fast Fourier Transform (FFT) to analyze traffic scenes. The multi-layer fusion dynamically computes temporal dependencies and updates correlations between visual features. The framework has been evaluated on three real-world datasets and has shown robust performance, especially in challenging scenarios with limited training data.

**Strengths:**

1. Clarity of Writing: The paper is well-written with a clear and logical flow, making it easy to understand the proposed system and its components.
2. Visualization: The figures and tables are clearly designed, enhancing the readability and comprehension of the framework and results.
3. Dataset Validation: The authors have validated their model on multiple datasets, which speaks to the robustness and generalizability of the CRASH framework.

**Limitations:**

1. Innovation: The paper presents a framework that shares significant similarities with prior work, such as DSTA. The authors could improve the novelty by providing a more detailed motivation and a comparative analysis that highlights the unique contributions and advantages of CRASH.
2. Framework Description: The abstract lacks a clear motivation for the study. A more compelling rationale for the development of CRASH over existing methods would strengthen the paper.
Figure Captioning: The caption for Figure 1 could be more descriptive to better explain the framework's architecture without needing to refer back to the text.
3. Ablation Studies: The analysis of ablation studies is not as thorough as it could be. While Table 6 contains valuable data, the discussion in the text (743-784) lacks detailed description and quantitative analysis to fully understand the impact of each component.
4. Detail Issues: There are minor issues such as a citation format error on line 152 and inconsistency in the use of abbreviations for "Table" and "Fig.", which should be corrected for publication standards.
Given the strengths and limitations outlined, my recommendation is borderline reject. The paper presents a solid framework with good empirical results but needs to address the issues of novelty and detailed analysis to be considered for acceptance. The final score may be adjusted based on the authors' rebuttal and their ability to address these concerns.

**Suitability:**

2

---

### Official Review · Reviewer_oLGU · 2024-05-19

**Rating:** 5
**Confidence:** 3

**Summary:**

The paper presents CRASH, an innovative framework designed to enhance the collision prediction capabilities of autonomous vehicles (AVs), a critical advancement for improving safety. Addressing the inherent unpredictability and complexity of traffic accidents, the CRASH system integrates a suite of components including an object detector, feature extractor, object-aware module, context-aware module, and multi-layer fusion. Notably, the object-aware module prioritizes high-risk objects through spatial-temporal analysis, while the context-aware module broadens visual information using FFT. The multi-layer fusion dynamically assesses temporal dependencies for precise predictions. Tested on real-world datasets, CRASH outperforms current benchmarks in Average Precision (AP) and mean Time-To-Accident (mTTA), showcasing robustness and adaptability, particularly in scenarios with limited training data. This work significantly contributes to the field by pushing forward the performance boundaries of collision prediction for AVs.

**Strengths:**

Innovation: The paper introduces an innovative accident prediction framework termed "CRASH," which enhances the accuracy of accident anticipation by integrating global contextual information with intricate spatio-temporal interactions.

Theoretical Approach: The authors have developed a novel Object Focus Attention (OFA) mechanism and a context-aware module that leverages the Fast Fourier Transform (FFT) and Context-aware Attention Blocks (CAB). These innovations allow for the computation of fine-grained correlations between nuanced spatial and appearance changes in different objects, contributing to a deeper understanding of the dynamics preceding an accident.

Experimental Design and Data Augmentation: The paper's experimental design is robust, with comprehensive benchmark tests conducted on the DAD, A3D, and CCD datasets. The authors have demonstrated the superior performance of the CRASH framework over state-of-the-art (SOTA) baselines across key metrics. A particular strength of this work is the innovative approach to data augmentation, simulating the variability and randomness of missing data encountered in real-world scenarios. This provides a more realistic evaluation of the model's performance.

Overall, the manuscript is well-structured, with a clear presentation of ideas and findings. The logical progression of the paper is easy to follow.

**Limitations:**

The paper contributes to the field of autonomous vehicle safety with the CRASH framework, which integrates several novel components for accident anticipation. However, there are a few aspects that require further consideration:

Computational Efficiency: The integration of FFT and multi-layer fusion in the CRASH framework is an innovative approach; however, it may come at the cost of significant computational overhead. This could result in substantial processing delays, which is a critical concern in the context of real-time collision prediction for autonomous vehicles. If the computational demands lead to latency, the system might fail to predict collisions in time. It is suggested that the paper include an analysis of the computational efficiency.

Object Detection Accuracy: A potential limitation is the framework's dependence on object detection, where errors could significantly affect collision predictions. The paper would benefit from a sensitivity analysis on how object detection inaccuracies impact the system's reliability.

**Suitability:**

2

---

### Official Review · Reviewer_hStj · 2024-05-24

**Rating:** 2
**Confidence:** 4

**Summary:**

The article introduces a novel framework called CRASH, designed for predicting traffic accidents around autonomous vehicles (AVs) using camera footage, to enhance AV safety. The main challenges include the unpredictable nature of traffic accidents, their long-tail distribution, the complexity of traffic scene dynamics, and the limited field of vision of onboard cameras.

The CRASH framework integrates five components: an object detector, a feature extractor, an object-aware module, a context-aware module, and multi-layer fusion. The object-aware module prioritizes high-risk objects by calculating the spatiotemporal relationships between traffic agents. The context-aware module extends visual information from the temporal to the frequency domain using Fast Fourier Transform (FFT), capturing detailed visual features and broader context cues. The multi-layer fusion dynamically computes temporal dependencies between different scenes, iteratively updating correlations between visual features for accurate and timely accident prediction.

Evaluated on real-world datasets (DAD, CCD, and A3D), the CRASH model surpasses existing top baselines in key evaluation metrics such as Average Precision (AP) and mean Time-To-Accident (mTTA). Its robustness and adaptability are particularly evident in challenging driving scenarios with missing or limited training data, demonstrating significant potential for application in real-world autonomous driving systems.

**Strengths:**

1. The collaboration in this paper adheres to academic standards.
2. A considerable number of comparative experiments were conducted, and some results show noticeable improvements.
3. The figures are clear and aesthetically pleasing.

**Limitations:**

1. This paper only considers a single camera as input, which does not align with the multimodal theme of MM24.
2. Current mainstream autonomous vehicles are equipped with multi-view cameras and LiDAR covering 360 degrees, and LiDAR is crucial for perception. However, this paper only explores the use of a single camera. Additionally, for collision prediction, it is essential to consider displacement and velocity information obtained from IMU and GNSS, as vehicles are not always stationary or moving in a straight line at a constant speed. Therefore, the problem chosen in this paper lacks sufficient practical value.
3. The lack of code makes it difficult for reviewers to replicate the results. Additionally, there is no plan to release the code, which reduces the contribution to the community.
4. The motivation is unclear, and there is no solid theoretical basis to justify the necessity and rationality of the proposed method. It seems more like finding a target after shooting an arrow.
5. There are some grammatical issues, such as a question mark in the citation on line 152.

**Suitability:**

1

---

### Official Review · Reviewer_31c2 · 2024-05-24

**Rating:** 4
**Confidence:** 3

**Summary:**

This paper introduces CRASH that is a framework for autonomous vehicle (AV) accident anticipation. This framework deals with unpredictable traffic, uncommon accident occurrences, complex scene dynamics, and restricted camera views etc. It combines five components: object detection, feature extraction, an object-aware module to prioritize high-risk objects, a context-aware module to capture broader scene information using the Fast Fourier Transform (FFT), and a multi-layer fusion module that dynamically analyzes temporal connections between scenes and visual features. The authors show that CRASH performs better than existing models on real-world datasets, achieving high average precision (AP) and mean time-to-accident (mTTA) measurements. The experiments are conducted with real-world dataset, including Dashcam Accident Dataset (DAD), Car Crash Dataset (CCD), and AnAn Accident Detection (A3D) datasets.

**Strengths:**

The paper is well structured with a clear presentation of the research on accident prediction.

The experiments are conducted with three datasets.

The proposed methods showcase good results on different datasets with complete or missing data.

**Limitations:**

The five-component architecture (object detection, feature extraction, object-aware module, context-aware module, multi-layer fusion) raises questions about its necessity for accident classification and time-to-accident (TTA) prediction. A deeper understanding of the relationships between these modules and their individual contributions to the overall performance is crucial. The current ablation study only analyzes the impact of removing individual components, but a more thorough examination is needed to fully grasp their interplay within the framework.

The whole model has VGG and C-RCNN for object detection. Exploring other object detection models could potentially improve performance or offer a more efficient solution.

The paper mentions using VGG16 backbones for both object detection and feature extraction.  It's unclear whether these backbones are shared or separate.  Furthermore, the model differentiates object-aware and context-aware features based on shape, but a more detailed analysis of their feature distributions and how they differ is missing. For exmaple, visualization of these features would be helpful in understanding their characteristics.

The proposed object-aware module requires further explanation to differentiate it from a standard self-attention module.  Specifically, how does the attention mechanism focus on objects relevant to accident anticipation?

Similar to the object-aware module, a comparison between the proposed context attention block and a general self-attention block is necessary. Understanding the structural differences would be beneficial.

The paper mentions that the Fast Fourier Transform (IFFT) and the context attention block improve the training stability of the model. However, the reviewer suggests that this claim needs further explanation and justification.

Table 1 shows good performance on complete datasets, but the significant increase in complexity compared to previous work doesn't seem to be fully justified by the marginal performance improvement.

Minor: Figure 1 lacks the input to the context-aware module, hindering a complete understanding of its functionality.

**Suitability:**

3

---

### Meta-Review · Area_Chair_nKCv · 2024-07-10

**Recommendation:** Accept (Poster)
**Confidence:** 4

**Metareview:**

This paper proposes a method to anticipate future crashes integrating information from object detectors. While some reviewers raised concerns the rebuttal is very thorough even adding additional ablation studies and clarifying novelty. If this modification can be added in the final version the paper can be considered solid.